# Coexisting Conditions Modifying Phenotypes of Patients with 22q11.2 Deletion Syndrome

**DOI:** 10.3390/genes14030680

**Published:** 2023-03-09

**Authors:** Marta Smyk, Maciej Geremek, Kamila Ziemkiewicz, Tomasz Gambin, Anna Kutkowska-Kaźmierczak, Katarzyna Kowalczyk, Izabela Plaskota, Barbara Wiśniowiecka-Kowalnik, Magdalena Bartnik-Głaska, Magdalena Niemiec, Dominika Grad, Małgorzata Piotrowicz, Dorota Gieruszczak-Białek, Aleksandra Pietrzyk, T. Blaine Crowley, Victoria Giunta, Daniel E. McGinn, Elaine H. Zackai, Oanh Tran, Beverly S. Emanuel, Donna M. McDonald-McGinn, Beata A. Nowakowska

**Affiliations:** 1Department of Medical Genetics, Institute of Mother and Child, 01-211 Warsaw, Poland; 2Institute of Computer Science, Warsaw University of Technology, 75, 00-662 Warsaw, Poland; 3Department of Genetics, Polish Mother’s Memorial Hospital Research Institute, 70-445 Łódź, Poland; 4Department of Medical Genetics, Children’s Memorial Health Institute, 04730 Warsaw, Poland; 5Department of Genetics and Pathology, Pomeranian Medical University, 70-204 Szczecin, Poland; 6Division of Human Genetics and 22q and You Center, the Children’s Hospital of Philadelphia, Philadelphia, PA 19104, USA; 7Perelman School of Medicine of the University of Pennsylvania, Philadelphia, PA 19104, USA

**Keywords:** 22q11.2DS, genomic disorder, copy number variation, CNV, single nucleotide variant, SNV, array, exome sequencing

## Abstract

22q11.2 deletion syndrome (22q11.2DS) is the most common genomic disorder with an extremely broad phenotypic spectrum. The aim of our study was to investigate how often the additional variants in the genome can affect clinical variation among patients with the recurrent deletion. To examine the presence of additional variants affecting the phenotype, we performed microarray in 82 prenatal and 77 postnatal cases and performed exome sequencing in 86 postnatal patients with 22q11.2DS. Within those 159 patients where array was performed, 5 pathogenic and 5 likely pathogenic CNVs were identified outside of the 22q11.2 region. This indicates that in 6.3% cases, additional CNVs most likely contribute to the clinical presentation. Additionally, exome sequencing in 86 patients revealed 3 pathogenic (3.49%) and 5 likely pathogenic (5.81%) SNVs and small CNV. These results show that the extension of diagnostics with genome-wide methods can reveal other clinically relevant changes in patients with 22q11 deletion syndrome.

## 1. Introduction

22q11.2DS is the most common chromosomal microdeletion syndrome in humans, with a prevalence of 1:2148 live births [1,2]. The morbidity in utero is higher with the frequency of 1:1000 fetuses [3,4] and 1:1500 miscarriage samples [5]. 22q11 deletion is a result of non-allelic homologous recombination between high identity low copy repeats (LCRs). Most of the affected individuals (~90%) carry a typical hemizygous 3 Mb deletion on chromosome 22q11.2, mediated by LCRs 22-A and 22-D, but smaller deletions mediated by other combinations of LCRs are also described [6]. This leads to a diminished dosage of nearly 60 genes, including coding and noncoding RNAs.

The characteristics of this contiguous gene deletion syndrome is highly variable phenotypic severity with many common significant physical and behavioral clinical findings described, as well as numerous rare significant features impacting clinical care, but none appear to be fully penetrant and each exhibits variable expressivity [2,7,8]. Variability is observed across unrelated patients and between affected family members who have inherited identical deletions, including amongst identical twins [9,10,11,12]. Approximately 10% of chromosome 22q11.2 deletions are parentally transmitted, and in those cases, high penetrance and wide expressivity are observed [11,13,14]. Frequently associated features of 22q11.2DS are congenital heart defects, palatal differences, immunodeficiency, endocrine abnormalities, developmental delay, cognitive deficits, and behavioral phenotypes [7,8,14,15]. Less commonly observed are structural ophthalmologic abnormalities, choanal atresia, hearing loss, laryngo-tracheo-esophageal abnormalities, gastrointestinal differences, congenital diaphragmatic hernia, vertebral anomalies, polydactyly, IUGR, idiopathic seizures, microcephaly, neural tube defects, ADHD, and autism [14]. Patients with 22q11.2DS have been reported to have a diminished adult life expectancy, with an increased risk of sudden death and developing psychiatric disorders [16,17]. 

It remains unknown why individuals with deletion of the same size present such a wide range of phenotypes. Some of the genes located within the standard deletion have major clinical impact, particularly T-box 1 (*TBX1*). The *TBX1* gene is part of the larger family of T-box genes, which help to regulate tissue and organ formation during development. However, a different minority of patients harbor nested distal deletions but retain two copies of the *TBX1* gene [2]. Therefore, the single dose of deleted genes alone cannot explain the tremendous variation in the severity and penetrance of the associated clinical features among affected individuals. Differences in deletion extents also cannot account for the huge variability in phenotypic severity among patients with 22q11.2DS [18]. Currently, there is no association between deletion size and the presence of the most common clinical symptoms, congenital heart disease or palate anomalies [19], although the common clinical features are less penetrant in smaller atypical nested deletions [14]. This points out that genetic variants, beyond the deleted genes themselves, may play a role in determining the 22q11.2DS clinical complications. Therefore, sequencing and microarray methods can help uncover, at least partially, genetic differences among patients with 22q11.2DS that can influence disease severity.

Several potential genetic mechanisms can underlie these highly variable phenotypic manifestations, including that they may be attributed to the hemizygous variants on the remaining allele, due to additional rare pathogenic variants elsewhere in the genome, as well as nongenetic environmental factors. Unmasking autosomal recessive disorders by having a deletion in one 22q11.2 allele and a mutation on the other non-deleted allele has been described in *GP1BB* (Bernard–Soulier syndrome type B) [20], *SNAP29* (CEDNIK syndrome—Cerebral dysgenesis, neuropathy, ichthyosis, and keratoderma) [21], *LZTR1* (Noonan syndrome) [22,23], *SCARF2* (van den Ende-Gupta syndrome (VDEGS) [24], and *CDC45* (CGS syndrome—craniosynostosis, gastrointestinal differences, short stature, and skeletal differences/Meier–Gorlin syndrome/Baller–Gerold syndrome) genes [25,26]. Mapping the variation in the remaining alleles in patients with 22q11.2DS also identified potential variants in *PI4KA* and *PRODH* [27]. Recently, known pathogenic genetic factors beyond the 22q11.2 region, including mutations and chromosomal structural aberrations, were described in ~0.9% of patients with 22q11.2DS with atypical phenotypic features [28]. 

Our goal was to determine the presence of additional variants affecting the phenotype of patients with 22q11.2DS. 

## 2. Materials and Methods

### 2.1. Patients and Deep Phenotyping

225 patients were enrolled in the study after receiving genetic counselling and signing informed consent. The study group included 82 prenatal cases and 143 postnatal patients. During routine aCGH diagnostics at the Institute of Mother and Child, we identified 159 patients with 22q11.2 deletion (82 prenatal and 77 postnatal). For exome sequencing, 86 postnatal patients (40 females and 46 males) were recruited/evaluated at the Department of Medical Genetics, Institute of Mother and Child, Warsaw, Poland (76 probands), and within the 22q and You Center, The Children’s Hospital of Philadelphia, Philadelphia, PA, USA (10 probands). 20 patients were analyzed by both methods: array CGH and exome sequencing. In the remaining 66 patients, the presence of a chromosome 22q11.2 deletion had been identified by FISH or MLPA. 

To obtain the most detailed clinical description of the patients in each case, medical consultation was carried out by experienced clinical geneticists. A standard form (Appendix A) for deep phenotyping was prepared at the Institute of Mother and Child on the basis of review of the literature and included clinical features that may occur in individuals with 22q11.2DS. The clinical symptoms were grouped by category: cardiovascular system, immunodeficiency, autoimmune disease, palatal abnormalities/velopharyngeal insufficiency/dysfunction, laryngo-tracheo-esophageal anomalies, endocrinologic problems including hypocalcemia, short stature, intrauterine growth deficiency/failure to thrive (FTT), macrocephaly, microcephaly, structural eye defects and ocular disorders, skeletal abnormalities, dental problems, structural ear defects and hearing problems, structural CNS anomalies, unprovoked seizures, genitourinary tract anomalies, inguinal hernia, gastrointestinal anomalies and functional problems, umbilical hernia, craniofacial dysmorphic features, psychomotor and intellectual development, learning disabilities/cognitive deficits, behavioural abnormalities, and psychiatric illness. A detailed form with over 230 clinical features was completed where possible (Appendix A). The research was approved by the Bioethics Committee of the Institute of Mother and Child in Warsaw and the Institutional Review Board of the Children’s Hospital of Philadelphia.

### 2.2. aCGH

We performed array comparative genomic hybridization (arrayCGH), with commercially available arrays such as the 60K CytoSure Constitutional v3 microarray (Oxford Gene Technology, Oxford, UK) according to manufacturer’s instructions. CNVs that showed partial or complete overlap with known segmental duplications were excluded from further analysis, due to the high variability of copy number variations in those regions. CNVs were classified as pathogenic, likely pathogenic, likely benign, benign and of uncertain significance (VUS) based on clinical data in known CNV databases: Database of Genomic Variants (http://dgv.tcag.ca/dgv/app/home accessed on 18 November 2022), ClinVar (https://www.ncbi.nlm.nih.gov/clinvar/ accessed on 18 November 2022), DECIPHER (https://www.deciphergenomics.org/ accessed on 18 November 2022), ClinGen (https://www.clinicalgenome.org/ accessed on 18 November 2022). Recurrent CNVs or CNVs associated with known microdeletion or microduplication syndromes were classified as pathogenic or likely pathogenic depending on the penetrance and clinical features present in probands. In 159 patients (82 prenatal and 77 postnatal), array CGH was performed as a first diagnostic test. Additionally, we conducted aCGH where exome sequencing revealed CNV, which could be confirmed (due to size and array coverage).

### 2.3. Exome Sequencing

Exome sequencing (ES) was applied to identify rare variants that could affect the phenotype in 86 patients with 22q11.2DS. Detailed phenotyping of all patients was performed (Appendix A) to estimate the correlation between ES results and specific phenotypes. 

Sequencing was performed using Illumina HiSeq (San Diego, CA, USA) instruments after exome capture with the Sure Select All Human V6 design. Raw sequence data were post-processed using the bc-bio pipeline (https://github.com/chapmanb/bcbio-nextgen accessed on 18 November 2022). The bc-bio pipeline performs mapping of the short reads against a human genome reference sequence (GRCh38) using the Burrows-Wheeler Alignment (BWA), BAM post-processing using GATK, and variant calling using the GATKHaplotype Caller. Finally, ANNOVAR [29] was used to annotate relevant information about gene names, predicted variant pathogenicity, reference allele frequencies and metadata from external resources, and then to add these to the Variant Call Format (VCF) file. 

Next, we used HMZDelFinder algorithm [30] to search for small, hemizygous deletions within the 22q11.2 region and small, heterozygous deletions beyond 22q11.2 region. As a control data set, we used WES data from 164 samples sequenced at the same platform and processed using the same pipeline, as our patients with 22q11.2DS. To prepare input data for HMZDelFinder algorithm, for each sample for every exonic target from the capture design we computed RPKM values, i.e., number of reads per kilobase of transcript per million mapped reads. To detect heterozyous CNVs outside of the 22q11.2 region we used CoNIFER tool with its default parameters [30]. As an input we reused the same rpkm data previously precomputed to run HMZDelFinder. Each putative CNV identified by CoNIFER has been visualized and undergone manual inspection to remove common polymorphisms and likely false positive calls.

## 3. Results

### 3.1. aCGH Analysis

In 21 patients, in addition to the chromosome 22q11.2 loss, we identified 31 copy number variants. Among those 31 CNVs, 5 were pathogenic, 5 were likely pathogenic, and 21 CNVs were classified as variants of uncertain significance (Table 1). In 8 patients, more than one additional CNV was observed. Ten out of thirty-one additional CNVs also occurred on chromosome 22q, but in different regions, however in 7 of those cases the additional deleted regions most likely were the extension of the typical or nested 22q11. Importantly, these variants would not be possible to identify using targeted methods, like FISH or MLPA. 

### 3.2. Exome Sequencing Analysis

#### 3.2.1. Variants in Genes from Remaining 22q11.2 Region

Hemizygotic variants in the 22q11.2 deletion region were analyzed with a frequency of 0.01 in gnomAD database. Next, only exonic, stopgain, nonsynonymous, and frameshift variants were taken into analysis. Across all 86 patients, a total of 29 variants in 21 patients were identified in the remaining 22q11.2 region (Appendix A). None of the variants were previously described as pathogenic or likely pathogenic, therefore all variants were classified as variants of uncertain clinical significance. However, investigation of the remaining allele in our tested cohort revealed a rare nonsynonymous-damaging variant in *CDC45* in one patient out of 8 with craniosynostosis. In addition to bilateral coronal craniosynostosis, this patient (GC028958) also presented with additional clinical features overlapping with the previously described MGORS7/CGS syndrome, such as an anteriorly placed anus.

HMZDelFinder identified two putative hemizygous deletions in the remaining 22q11.2 region. The first deletion encompasses two genes; *DGCR6*, *PRODH* (Chr22:18906447-18931248). This deletion is classified in ClinVar as likely benign and had no effect on the patient’s phenotype, because the region is commonly deleted in control populations. However, this deletion was never described in patients with 22q11.2DS. Therefore, we cannot exclude the possibility that nullisomy of *PRODH* has an impact on the patient’s long-term phenotype. This deletion was classified as uncertain clinical significance. The second deletion (hemizygous), also identified in the 22q11.2 region, encompasses 8 exons of *TANGO2* (chr22:20043265-20064650). Bi-allelic truncating mutations in this gene have been associated with TANGO2-related metabolic encephalopathy and arrhythmias presenting with recurrent muscle weakness with rhabdomyolysis, metabolic crises, and cardiac arrhythmia [31]. Therefore, this variation was confirmed by a qPCR method, classified as pathogenic, and was consistent with the patient’s phenotype (Figure 1). 

#### 3.2.2. Filtering for SNVs Elsewhere in the Genome

Filtering for known pathogenic variants, defined as a frequency less than or equal to 0.001 in gnomAD genome database, autosomal dominant inheritance, and classified as pathogenic in ClinVar database, revealed 1 rare stopgain, 1 rare frameshift deletion, 1 rare UTR3 variant, and 4 rare non-synonymous variants. In total, 6 genes across 86 patients carry a rare protein-altering variant. The clinical outcome of these variants was discussed with clinicians for all individuals (Table 2). For two cases, the clinical features were consistent with the predicted consequences of genomic variant and were classified as pathogenic. In five cases, the patients did not present expected phenotype features, and therefore the variants have been classified as potentially pathogenic. 

Rare SNVs were selected using filtering with an allele frequency less than 0.001 in gnomAD and were then filtered against an in-house control dataset. Variants in genes associated with OMIM diseases, of autosomal and X-linked dominant inheritance or X-linked recessive inheritance in male patients, were chosen. Effects of missense variants were predicted using four in silico analysis tools (SIFT; Polyphen2_HDIV; Polyphen2_HVAR; MutationTaster). SNVs with deleterious predictions in at least 2 out of 4 prediction tools or SNV automatically deleterious in Mutation Taster were selected. Next, phenotypic features of patients with those SNVs were compared with clinical features related to the implicated genes. Only those variants, where the clinical features overlapped with predicted phenotypes, were included in the table (Table 3). The pathogenicity of all selected SNVs was estimated using Alamut-2.11-0 software. 

HMZDelFinder revealed three heterozygous deletions in two genes: GABRA5 and KARS (Table 4). 

Additionally known cancer susceptibility SNVs in genes associated with cancer risk have been identified in 5 patients (Appendix A). The variants have been revealed in 3 genes: ATM, BRCA1, and BRCA2. All patients were informed regarding these risk genes and were referred to the oncology clinic.

Overall, 885 variants of uncertain clinical significance were selected based on the frequency (less than 0.0001 in gnomAD database), function (exonic), and type of change (frameshift deletion, frameshift insertion, stopgain, or nonsynonymous (Appendix A). 

In summary, we have identified 8 pathogenic variants, and 10 likely pathogenic variants in 225 patients, which is much more than previously expected (Table 5). 

## 4. Discussion

In our study, we have analyzed genomes of 225 patients with 22q11.2 deletion syndrome, using array CGH and/or exome sequencing methodology. The aim of our research was to identify variants beyond the 22q11.2 deletion, which may contribute to the clinical complexity in our patients. 

Hemizygous variants are one of the potential genetic mechanisms that could underlie this clinical variability. Allelic variations of genes within a critical region of the non-deleted chromosome can have an impact on phenotypes by unmasking recessive diseases, which has already been proven for several genes within the 22q11.2 region, like *SNAP29* and *GP1BB* [21,32]. Recent studies have mainly focused on mapping mutations on the remaining allele [27,33]. This strategy has demonstrated high efficiency in cases with atypical and less common clinical features. For example, biallelic mutations in *CDC45* are thought to be the cause of craniosynostosis in Meier–Gorlin syndrome (MGORS7, OMIM 617063) [25], a rare autosomal recessive primordial dwarfism disorder, characterized by microtia, short stature, and absent or hypoplastic patellae. Recently, in patients with 22q11.2DS, the alteration of *CDC45* was associated with the pathogenesis of craniosynostosis, as well as phenotypic features more frequently reported in association with Baller–Gerold (OMIM 218600) and RAPADILINO syndrome (OMIM 266280), as well as unique previously-unreported features [26]. In fact, rare hemizygous variants in the *CDC45* gene were found in 5/15 (33%) of the patients reported by Unolt et al., with both a 22q11.2 deletion and atypical clinical features in 3/7 of the patients (43%) including craniosynostosis (2 bicoronal, 1 metopic) and none of the 133 22q11.2DS patients from the control group (Fisher’s test *p* value < 0.01). These patients also had important structural anomalies not typically associated with 22q11.2DS, most significantly some combination of anal/intestinal anomalies, limb differences, short stature, and craniofacial anomalies. Investigation of the remaining allele in our tested cohort revealed a rare nonsynonymous damaging variant in *CDC45* in one patient out of eight with craniosynostosis. In addition to bilateral coronal craniosynostosis, this patient also presented other clinical features overlapping with MGORS7/CGS, such as an anteriorly placed anus. 

In two patients, we found additional deletions on the remaining allele. In one patient, an intragenic deletion of 8 exons in *TANGO2* was identified following a postmortem study. Mutations in this gene are responsible for TANGO2-related metabolic encephalopathy and arrhythmias, an autosomal recessive recurrent metabolic encephalomyopathic crises, rhabdomyolysis, cardiac arrhythmia, and intellectual disability syndrome (OMIM 616878). A large proportion of patients described in the literature have succumbed in childhood due to cardiac arrest related to arrhythmias or following seizures related to a hypoglycemic crisis. Some patients, with variants in the *TANGO2* gene, presented primarily with neurologic features [34]. In our second patient, we identified a deletion encompassing the whole *PRODH* gene. Mutations or small deletions in *PRODH* cause recessive hyperprolinemia (OMIM 239500) with neurologic manifestations, including seizures. It is well known that individuals with 22q11.2DS are at high risk for developing schizophrenia and schizoaffective disorder. *PRODH* was suggested to play a role in the later stages of neural development [35]. Although the latest research does not support the theory that high proline level is responsible for psychosis, it cannot be excluded that nullisomy of the *PRODH* gene has no clinical effect. Despite the association between variants in *PRODH* and an increased risk of schizophrenia shown in some studies, the potential role of this gene in psychiatric diseases remains unclear [36,37]. 

Hemizygous variants on a second allele may explain only part of the clinical variability associated with 22q11.2DS. The phenotypic variability may also be associated with additional copy number variations, as was shown in our study. 

Analysis of other microdeletions with variable expressivity suggests that a two-hit model may be more generally applicable to neuropsychiatric disease [38]. Additional CNVs were already suggested to be genomic modifiers in patients with a 22q11.2 deletion or duplication [39,40]. In our study we have identified 31 additional CNVs in 21 patients, among 159 tested by array CGH. It is worth noting that more than one CNV was observed in 8 patients in our cohort, worsening the overall clinical picture. In 10 cases the additional CNV was located on chromosome 22q, however in 7 of those cases the additional deleted regions most likely were the extension of the typical or nested 22q11 deletions. It is particularly interesting that additional CNVs occurred in 7 patients with nested 22q11 deletions and 14 patients with A-D typical deletion. Taking into account the ratio of nested to the typical deletions (28:131 in our study), the result can suggest that if the smaller deletions have a milder phenotype than the classical ones, they may more likely have additional CNVs affecting phenotype. However, to draw final conclusions, more cases with different sizes of the deletion need to be studied. Moreover, in 3 cases, recurrent deletions with variable expressivity on chromosome 15 and 16 were revealed. Recurrent proximal 16p11.2 microdeletion found in case GC034796 can also influence some phenotypic features like global developmental delay, hypotonia, or cardiac anomalies. Rare CNVs outside of the 22q11.2 region may modify risk for congenital cardiac defects in some patients, with 22q11.2DS probably being due to the content of genes influencing heart development [41,42]. In patient GC034823, who demonstrated cognitive regression and dementia, both of which are unusual for 22q11.2DS, the deletion region is expanded and encompasses the *TUBA8* gene. Mutations in *TUBA8* were associated with recessive polymicrogyria [43]. Recently, Dantas et al. reported downregulation of this gene in patients with 22q11.2 deletion syndrome [44]. Another CNV encompassing gene associated with an intellectual disability is partial duplication of the *MID2* gene, which was revealed in male patient GC034808. Previously, a duplication overlapping *MID2* was reported in a boy with FG syndrome, an X-linked multiple congenital anomalies syndrome (OMIM 300581), who had hypotonia and developmental delay [45]. A missense mutation in *MID2* was described in a large Indian family with global developmental delay and minor facial features [46]. Recently, an Xq22 duplication including *MID2* was described in a patient whose phenotypic features were not consistent with FG syndrome [47]. Another mechanism modifying patients’ clinical features is the influence of single nucleotide variations (SNVs) in genes that reside outside of the deleted region, e.g., SNVs in genes that lie on the same pathway with genes from the deleted region can modify the severity of symptoms. A good example of such a mechanism demonstrates patient GC034900, with a variant in the *TLL1* gene. Heterozygous mutations in the *TLL1* gene are responsible for atrial septal defect (ASD6). In about 1% of patients with 22q11.2DS. another known pathogenic genetic factor, such as mutations, chromosomal structural aberrations. or mosaic aneuploidy, was observed, resulting in dual diagnoses [28]. An example of a dual diagnosis in our cohort is patient GC030951, with both 22q11.2DS and a pathogenic variant in the *NF1* gene resulting in Neurofibromatosis as well as 22q11.2DS.

Historically, medical evaluation of genetic conditions relies upon pattern recognition, which recognizes a phenotype, signs, or symptoms to guide targeted genetic testing. The clinical features of many genetic conditions and 22q11.2 deletion syndrome may overlap. Therefore, most often, when a 22q11.2 deletion is identified, further diagnosis is frequently not considered in most patients, and additional or rare clinical features are attributed to the presence of the deletion.

22q11.2DS is a multi-system disorder caused by a single deletion that is genetically complex and can be seen as a model for gene–gene interactions and phenotype–genotype correlations. Recent advances in genetics and systems biology provide excellent opportunities to gain novel insights into pathways underlying the morbidities associated with 22q11.2DS across the lifespan.

Our study demonstrates that secondary diagnoses should be taken into account in all patients with 22q11.2DS. Only a whole-exome/whole-genome study of large cohorts of patients with 22q11.2DS will provide a robust opportunity to determine genotype–phenotype correlations for individual genes with the 22q11.2 deletion and beyond. Knowing how often additional variants can modify the patient’s phenotype, it is recommended to use the aCGH/SNP microarray methodology in cases of suspected 22q11.2 deletion at minimum.

## Figures and Tables

**Figure 1 genes-14-00680-f001:**
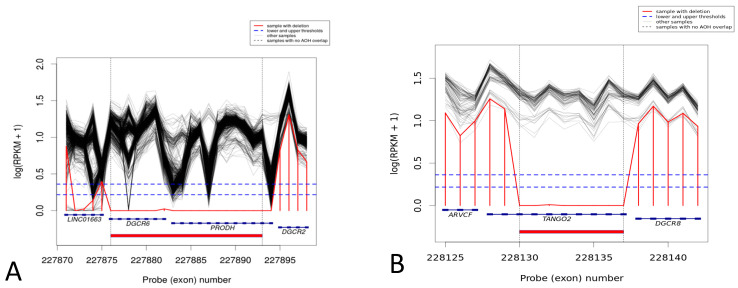
HMZDelFinder identified; (**A**) clinically uncertain, hemizygous deletion, in the remaining 22q11.2 region. The variant encompasses two genes: *DGCR6*, *PRODH* (Chr22:18906447-18931248). (**B**) Pathogenic, hemizygous deletion, in the remaining 22q11.2 region. The variant encompasses 8 exons of *TANGO2* (chr22:20043265-20064650).

**Table 1 genes-14-00680-t001:** Copy number variants identified in 159 (82 prenatal cases and 77 postnatal patients) patients, in addition to the 22q11 deletion. CNVs were classified as pathogenic, likely pathogenic, and of uncertain significance (VOUS) based on clinical data in known, free-access CNV databases. OMIM morbid genes are bolded.

Patient ID	Variant	Sex	Age	Size	Pathogenicity/ CNV Classification	Protein Coding Genes or Known CNV Region	Patients’ Phenotype	22q11.2 Recurrent Region
PD5864	arr[GRCh37] 11q24.2q25(125857822_134868420)x1	M	17–18 Hbd	9.01 Mb	pathogenic	38 protein coding genes, 11q (Jacobsen syndrome) region	Fetal death, lip and cleft palate, edema.	proximal, A-B
	arr[hg19] 22q11.1q11.21(16940617_18706059)x1			1.77 Mb	likely pathogenic	*CCT8L2*, *XKR3*, *GAB4*, ***IL17RA***, *TMEM121B*, *HDHD5*, ***ADA2***, *CECR2*, *SLC25A18*, ***ATP6V1E1***, *BCL2L13*, *BID*, *MICAL3*, ***PEX26***, ***TUBA8***, ***USP18***		
PD8038	arr[GRCh37] 22q11.21q11.23(21759521_23822984)x1	M	17–18 Hbd	2.06 Mb	pathogenic	22q11.2 recurrent region (distal type I, D-E/F)	Symmetrical hypotrophy of the fetus, cardiac defect	proximal, B-D
GC034796	arr[GRCh37] 13q21.32(67204211_67215612)x1	M	1 y 3 m	11 kb	uncertain clinical significance	exon 4 of *PCDH9*	Global developmental delay, ventricular septal defect, interrupted aortic arch, bicuspid aortic valve, atrial septal defect, aberrant subclavian, thymic aplasia with absent T cells, hypocalcemia, nasal regurgitation, thyroid hypoplasia, hypothyroidism, growth hormone deficiency, growth delay, dysmorphic features (micrognathia), scoliosis, butterfly vertebrae, additional ribs, hypotonia, ligamentous laxity, delayed dental eruption, chronic constipation	proximal, A-C
	arr[GRCh37] 16p11.2(29673967_30190593)x1			517 kb	pathogenic	16p11.2 recurrent region (proximal, BP4-BP5) (includes ***TBX6***)		
	arr[GRCh37] 22q11.21(18375151_18661758)x1			287 kb	uncertain clinical significance	exons 1–11 of *MICAL3*, ***PEX26***, ***TUBA8***, ***USP18***		
113787	arr[GRCh37] 20p12.3(8256689_8558204)x1	F	5 y	301 kb	uncertain clinical significance	*exon 3 of PLCB1*	Global developmental delay, delay of speech development, dysmorphic features	proximal, A-B
	arr[GRCh37] 22q11.1q11.21(16940617_18848020)x3			1.91 Mb	pathogenic	22q11.21 recurrent (Cat eye syndrome) region (includes *CECR2*)		
GC034823	arr[GRCh37] 22q11.1q11.21(17397633_18661758)x1	F	25 y	1.26 Mb	likely pathogenic	*GAB4*, ***IL17RA***, *TMEM121B*, *HDHD5*, ***ADA2***, *CECR2*, *SLC25A18*, ***ATP6V1E1***, *BCL2L13*, *BID*, *MICAL3*, ***PEX26***, ***TUBA8***, ***USP18***	Ventricular septal defect, double outlet right ventricle, pulmonary (valve) stenosis, unilateral cleft palate and lip, bifid uvula, velopharyngeal insufficiency, growth delay, dysmorphic features, myopia, scoliosis, dental problems, conductive hearing loss, nasopharyngeal reflux, chronic constipation, global developmental delay, delayed speech and language development, moderate intellectual disability, mood changes	proximal, A-B
	arr[GRCh37] 22q13.33(51162483_51178213)x3			16 kb	uncertain clinical significance	*ACR*, exons 1–21 of ***SHANK3***		
111183	arr[GRCh37] 16p12.2(21959950_22407951)x1	F	1 m	448 kb	likely pathogenic	16p12.2 recurrent region (proximal) (includes *EEF2K*, *CDR2*)	Facial dysmorphic features, cardiac defect, urinary system defect	proximal, A-D
105620	arr[GRCh37] 12q24.33(132239944_133773393)x1,	M	6 y	1.53 Mb	likely pathogenic	*SFSWAP*, *MMP17*, *ULK1*, ***PUS1***, *EP400*, *DDX51*, *NOC4L*, *GALNT9*, *FBRSL1*, *LRCOL1*, ***P2RX2***, ***POLE***, *PXMP2*, *PGAM5*, ***ANKLE2***, *GOLGA3*, *CHFR*, *ZNF605*, *ZNF26*, *ZNF84*, *ZNF140*, *ZNF891*, *ZNF10*, *ZNF268*	Hypotonia (low muscle tone), dysmorphic features, lack of speech development, cardiac defect	proximal, A-B
	arr[GRCh37] 22q11.1q11.21(17397633_18661758)x1			1.26 Mb	likely pathogenic	*GAB4*, ***IL17RA***, *TMEM121B*, *HDHD5*, ***ADA2***, *CECR2*, *SLC25A18*, ***ATP6V1E1***, *BCL2L13*, *BID*, *MICAL3*, ***PEX26***, ***TUBA8***, ***USP18***		
GC034824 /106502	arr[GRCh37] Xp11.23(47616035_48204099)x3	F	2 y 3 m	588 kb	uncertain clinical significance	***ZNF81***, *ZNF182*, *ZNF630*, *SPACA5*, *SPACA5B*, *SSX5*, *SSX1*	Aberrant subclavian artery, recurrent infections, thymic hypoplasia, hypernasal speech, nasal regurgitation, pronunciation defects, short stature, dysmorphic features (hypotelorism, bitemporal narrowing, micrognathia, retrognathia) long fingers, hypotonia, ligamentous laxity, dental problems, enuresis, gastrointestinal problems, global developmental delay	proximal, A-D
108894	arr[GRCh37] 1p36.23(8736229_9105539)x3 mat	M	2 y	369 kb	uncertain clinical significance	exons 1–2 of ***RERE***, *ENO1*, *CA6*, *SLC2A7*, exons 5–12 of *SLC2A5*	Global developmental delay, facial dysmorphic features, hypotonia (low muscle tone)	proximal, A-D
110226	arr[GRCh37] 4q13.1(65794809_66356976)x1 mat	M	11 y	562 kb	uncertain clinical significance	exons 1–2 of *EPHA5*	Hypotonia (low muscle tone), delay of speech development	proximal, A-D
111711	arr[GRCh37] 12q12(44202928_44445117)x1	M	1 m	242 kb	uncertain clinical significance	exons 1–3 of *TMEM117*	Facial dysmorphic features, cardiac defect	proximal, A-D
112367	arr[GRCh37] 22q11.22q11.23(23012069_23648159)x1 pat	M	9 m	637 kb	uncertain clinical significance	22q11.2 recurrent region (distal type II, E-F)	Cardiac defect	proximal, A-D
112587	arr[GRCh37] 15q13.2q13.3(30419801_32861612)x3	M	6 y	2.44 Mb	uncertain clinical significance	15q13.3 recurrent region (BP4-BP5) (includes *CHRNA7*)	No data	proximal, A-D
113324	arr[GRCh37] 7q21.3q22.1(97939915_98557740)x3 mat	F	1 y 4 m	618 kb	uncertain clinical significance	exons 1–8 of *BAIAP2L1*, *NPTX2*, *TMEM130*, exons 1–44 of *TRRAP*	Defect of the larynx, thymic hypoplasia, dysmorphic features	proximal, A-D
	arr[GRCh37] 22q11.23(23720181_25066484)x3 mat			1.35 Mb	uncertain clinical significance	22q11.2 recurrent region (distal type III, F-H)		
116123	arr[GRCh37] 18p11.31p11.23(7094706_8359012)x3	F	1 m	1.26 Mb	uncertain clinical significance	exon 1 of *LAMA1*, *LRRC30*, exons 1–23 of *PTPRM*	Facial dysmorphic features, defect of the urinary system, cardiac defect (vascular ring), long fingers.	proximal, A-D
PD2305	arr[GRCh37] 7p12.3p12.2(46094932_49190408)x3	M	22 Hbd	3.1 Mb	uncertain clinical significance	*TNS3*, ***PKD1L1***, *HUS1*, *SUN3*, *C7orf57*, *UPP1*, *ABCA13*, *CDC14C*	Screening test result showing increased risk of chromosomal aberration, fetal cardiac defect	proximal, A-D
PD3311	arr[GRCh37] 22q11.22q11.23(23012069_23648159)x1 pat	M	21 Hbd	636 kb	uncertain clinical significance	22q11.2 recurrent region (distal type II, E-F)	Fetal cardiac defect	proximal, A-D
GC034808	arr[GRCh37] Xq22.3(106791395_107156386)x2	M	6y	465 kb	uncertain clinical significance	*FRMPD3*, *NCBP2L*, ***PRPS1***, *TSC22D3*, exons 1–5 of ***MID2***	Recurrent infections, acute otitis media, submucosal cleft palate, bifid uvula, secondary hearing loss, pronunciation defects, speech articulation difficulties, short stature, dysmorphic features, hypotonia, ligamentous laxity, feeding difficulties, umbilical hernia, global developmental delay, delayed speech and language development, attention difficulties	proximal, A-D
108366	arr[GRCh37] 8p12(30498811_30732891)x3	M	9 y	234 kb	uncertain clinical significance	exons 1–2 of ***GTF2E2***, *SMIM18*, *GSR*, *UBXN8*, *PPP2CB*, exons 2–7 of ***TEX15***	Global developmental delay, facial dysmorphic features, epilepsy	proximal, A-B
GC034815	arr[GRCh37] 3p26.3(1539221_2206719)x1	F	17 y	667 kb	uncertain clinical significance	exons 1–2 of *CNTN4*	Patent foramen ovale, defects in humoral immunity, IgA deficiency, recurrent infections, acute otitis media, chronic sinusitis, T-cell lymphopenia, vitiligo, thrombocytopenia, Hashimoto thyroiditis, velopharyngeal insufficiency, laryngotracheoesophageal anomalies, hypothyroidism, hypoparathyroidism, dysmorphic features, myopia, amblyopia, hypotonia, ligamentous laxity, delayed dental eruption, feeding and swallowing problems, delayed speech and language development, supernumerary spleens	proximal, A-D
	arr[GRCh37] 7q21.3q22.1(97939915_98557740)x3			618 kb	uncertain clinical significance	exons 1–8 of *BAIAP2L1*, *NPTX2*, *TMEM130*, exons 1–44 of *TRRAP*		
GC028958	arr[GRCh37] 8q22.1(96815252_97229772)x1	F	2 weeks	414 kb	uncertain clinical significance	** *GDF6* **	Child succumbed at 14 days of life following her heart repair on day of life 13. Ventricular septal defect, persistent truncus arteriosus, atrial septal defect, bilateral coronal craniosynostosis, diffuse white matter gliosis, reduced cortical thickness, thymic aplasia with absent T cells, hypocalcemia, hypoparathyroidism, intestinal malrotation, anteriorly placed anus, diaphragmatic hernia	proximal, A-D
	arr[GRCh37] Xq28(151896983_152351582)x3			455 kb	uncertain clinical significance	*CETN2*, *CSAG1*, *MAGEA12*, *MAGEA2*, *MAGEA3*, ***NSDHL***, *PNMA3*, *PNMA5*, *PNMA6A*, *PNMA6C*, *ZNF185*		
	(8)x2~3			146.36 Mpz	pathogenic	mosaic trisomy		

**Table 2 genes-14-00680-t002:** Pathogenic and likely pathogenic SNVs, of autosomal dominant inheritance, were selected using filtering with an allele frequency less than 0.001 in gnomAD, then filtered against an in-house control dataset. EBD—Epidermolysis bullosa dystrophica.

ID	Sex	Age	Chr	Change	Zyg	Gene	Function	OMIM_Diseases Linked to Gene	Exonic Function	GnomAD Exome ALL	Patient’s Clinical Features Associated with Variant
GC034931	M	7 y	chr2	NM_000384:c.9115_9119del:p.F3039fs	het	*APOB*	exonic	Hypercholesterolemia, familial, 2; Hypobetalipoproteinemia	frameshift deletion	4.071 × 10^−6^	NA
GC034813	M	12 y	chrX	NM_000495:c.G1871A:p.G624D	hom	*COL4A5*	exonic	Alport syndrome 1, X-linked	nonsynonymous SNV	8.97 × 10^−5^	NA
GC034793	M	7 y	chr3	NM_000094:c.A425G:p.K142R	het	*COL7A1*	exonic	EBD inversa; EBD, Bart type; EBD, localisata variant; Epidermolysis bullosa dystrophica, AD; Epidermolysis bullosa dystrophica AR; Epidermolysis bullosa pruriginosa; Epidermolysis bullosa, pretibial; Toenail dystrophy, isolated; Transient bullous of the newborn	nonsynonymous SNV	4.468 × 10^−5^	NA
GC034931	M	7 y	chr11		het	*F2*	UTR3	Dysprothrombinemia; Hypoprothrombinemia; Thrombophilia due to thrombin defect		0	NA
GC030951	M	6 y	chr17	NM_000267:c.C5839T:p.R1947X	het	*NF1*	exonic	Leukemia, juvenile myelomonocytic; Neurofibromatosis, familial spinal; Neurofibromatosis, type 1; Neurofibromatosis-Noonan syndrome; Watson syndrome	stopgain	0	Neurofibromatosis
GC034900	F	1 y	chr4	NM_012464.5:c.713T>C:p.p.Val238Ala	het	*TLL1*	exonic	Atrial septal defect 6	nonsynonymous SNV	2 × 10^−4^	Atrial septal defect
GC034800	F	7 y	chr12	NM_000552:c.G2561A:p.R854Q	het	*VWF*	exonic	von Willebrand disease, type 1; von Willebrand disease, type 3; von Willebrand disease, types 2A, 2B, 2M, and 2N	nonsynonymous SNV	0.0034	NA

**Table 3 genes-14-00680-t003:** Rare SNVs of uncertain clinical significance. CNS—central nervous system; VSD—Ventricular septal defect; IAA—interrupted aortic arch; BAV—bicuspid aortic valve; ASD—atrial septal defect; PTA—patent ductus arteriosus; DORV—double outlet right ventricle.

Identifier	Sex	Age	Inheritance	Chr	Start	End	Ref	Alt	Zyg	Gene	OMIM Diseases	Patient’s Phenotype Features Correlating with OMIM Phenotype
GC028956	F	13 y	AD	chr9	128607934	128607934	A	C	het	*SPTAN1*	Developmental and epileptic encephalopathy 5	Structural CNS anomalies (spina bifida, polyhydramnios), unprovoked seizures, developmental delay
GC028956	F	13 y	AD	chr18	34794223	34794223	T	G	het	*DTNA*	Left ventricular noncompaction 1, with or without congenital heart defects	Congenital heart defects (VSD)
GC030952	F	24 y	AD	chr2	120982720	120982720	A	C	het	*GLI2*	Culler-Jones syndrome; Holoprosencephaly 9	Palatal abnormalities (bifid uvula), growth delay, delayed psychomotor development
GC034768	M	6 y	AD	chr19	10986246	10986246	G	A	het	*SMARCA4*	Coffin-Siris syndrome 4	Congenital heart defects (VSD, IAA, BAV, ASD), dental problems, hypotonia, global developmental delay, delayed speech and language development, learning disability
GC034778	F	10 y	AD	chr18	44952093	44952093	G	A	het	*SETBP1*	Intellectual developmental disorder, autosomal dominant 29; Schinzel–Giedion midface retraction syndrome	Global developmental delay, delayed speech and language development, learning disability
GC034779	F	7 y	AD	chr2	15945723	15945723	A	T	het	*MYCN*	Feingold syndrome 1	Asymmetric face, narrow palpebral fissures, epicantic folds, micrognathia, palatal abnormalities (bifid uvula), hearing loss, global developmental delay, delayed speech and language development, learning disability
GC034783	F	8 y	AD	chr1	211082791	211082791	G	A	het	*KCNH1*	Temple-Baraitser syndrome; Zimmermann-Laband syndrome 1	Cardiovascular system defects (ASD, conotruncal cardiac anomaly, aberrant subclavian), hypertelorism, skeletal abnormalities (proximal implantation of thumbs), learning disability
GC034785	M	19 y	AD	chr8	60800422	60800422	G	A	het	*CHD7*	CHARGE syndrome; Hypogonadotropic hypogonadism 5 with or without anosmia	Submucosal cleft palate, dysmorphic face (facial asymmetry, hypertelorism, malar flattening, micrognathia, cup ear), hearing loss, skeletal anomalies of limbs, feeding and swallowing problems, intellectual disability, learning disability, thymic hypoplasia, T-cell lymphopenia
GC034786	M	2 y 6 m	AD	chr1	7747776	7747776	A	G	het	*CAMTA1*	Cerebellar dysfunction with variable cognitive and behavioral abnormalities	Bulbous, wide nose, low-set ears, delayed speech and language development, hypotonia, dental problems, gastrointestinal problems
GC034787	M	1 y	AD	chr3	119402046	119402046	C	G	het	*ARHGAP31*	Adams-Oliver syndrome 1	Microcephaly, palatal anomalies (bifid uvula), cardiovascular system defects (VSD, PTA, Pulmonary artery stenosis), hypotonia, global developmental delay
GC034789	F	6 m	XLD	chrX	41134763	41134763	C	T	het	*USP9X*	Intellectual developmental disorder, X-linked 99, syndromic, female-restricted	Congenital heart defects (VSD, ASD), dysmorphic face (prominent forehead, bitemporal narrowing, posteriorly rotated ears, broad nasal bridge, bulbous nose), hypotonia, sensory processing problems
GC034796	M	1 y 3 m	AD	chr1	151440971	151440971	G	T	het	*POGZ*	White-Sutton syndrome	Congenital heart defects (VSD, ASD, IAA, BAV, aberrant subclavian), growth delay, dysmorphic face (low-set ears, posteriorly rotated ears, short philtrum), hypotonia, global developmental delay
GC034797	F	2 y 2 m	AD	chr2	227309265	227309265	C	T	het	*COL4A3*	Alport syndrome 3, autosomal dominant; Hematuria, benign familial	Sensorineural hearing loss
GC034800	F	7 y	AD	chr1	27552055	27552055	-	G	het	*AHDC1*	Xia-Gibbs syndrome	Low-set ears, global developmental delay, delayed speech and language development, intellectual disability, learning disability
GC034800	F	7 y	AD	chr6	45492054	45492054	C	T	het	*RUNX2*	Cleidocranial dysplasia; Cleidocranial dysplasia, forme fruste, dental anomalies only; Cleidocranial dysplasia, forme fruste, with brachydactyly; Metaphyseal dysplasia with maxillary hypoplasia with or without brachydactyly	Prominent forehead, dental problems, high-arched palate, skeletal abnormalities
GC034813	M	12 y	AD	chr16	3769345	3769345	C	A	het	*CREBBP*	Menke-Hennekam syndrome 1; Rubinstein-Taybi syndrome 1	Congenital heart defects (Tetralogy of Fallot, VSD, ASD, conotruncal heart defects) submucosal cleft palate, dysmorphic face (prominent forehead, broad nasal bridge, micrognathia, retrognathia, low-set ears), scoliosis, syndactyly, dental problems, cryptorchidism, recurrent infections, hypotonia, developmental delay, delayed speech and language development, intellectual disability
GC034813	M	12 y	XLD	chrX	45061396	45061396	T	C	hom	*KDM6A*	Kabuki syndrome 2	Congenital heart defects (conotruncal heart defects, Tetralogy of Fallot, VSD, ASD), submucosal cleft palate, cupped ears, dental problems, hypotonia, ligamentous laxity, developmental delay, behavioral difficulties
GC034822	M	10 y	AD	chr19	13300559	13300559	G	C	het	*CACNA1A*	Developmental and epileptic encephalopathy 42; Episodic ataxia, type 2; Migraine, familial hemiplegic, 1; Migraine, familial hemiplegic, 1, with progressive cerebellar ataxia; Spinocerebellar ataxia 6	Global developmental delay, hypotonia, intellectual disability (borderline), learning difficulties
GC034822	M	10 y	XLR	chrX	70454277	70454277	G	A	hom	*DLG3*	Intellectual developmental disorder, X-linked 90	Global developmental delay, delayed speech development, intellectual disability (borderline), learning difficulties
GC034824	F	2 y 3 m	AD	chr8	143728049	143728049	C	T	het	*FAM83H*	Amelogenesis imperfecta, type IIIA	Dental problems
GC034930	F	4 m	AD	chr3	111649747	111649747	A	C	het	*CD96*	C syndrome	Micrognathia, posteriorly rotated ears, epicantic folds, broad nasal bridge, short nose, congenital heart defect (VSD)
GC034930	F	4 m	AD	chr1	119916604	119916604	C	T	het	*NOTCH2*	Alagille syndrome 2; Hajdu-Cheney syndrome	Congenital heart defects (VSD, ASD, PTA), Bulbous nasal tip
GC034932	F	1 y	AD	chr12	115996599	115996599	C	T	het	*MED13L*	Impaired intellectual development and distinctive facial features with or without cardiac defects	Global developmental delay
GC034823	F	25 y	AD	chr8	105801714	105801714	G	A	het	*ZFPM2*	Tetralogy of Fallot	VSD, DORV with pulmonary stenosis
GC034789	F	6 m	AD	chr5	173233189	173233189	C	A	het	*NKX2-5*	Atrial septal defect 7, with or without AV conduction defects; Hypoplastic left heart syndrome 2; Hypothyroidism, congenital nongoitrous, 5; Tetralogy of Fallot; Ventricular septal defect 3	Congenital heart defects (ASD, VSD)
GC034808	M	6 y	AD	chr6	1610202	1610202	C	T	hom	*FOXC1*	Axenfeld-Rieger syndrome, type 3	Secondary hearing loss, redundant periumbilical skin

**Table 4 genes-14-00680-t004:** Pathogenic and of uncertain clinical significance deletions identified by HMZDelFinder.

ID	Sex	Age	Reults of HMZdelfinder Analysis	Gene Content	Gene Function
**pathogenic**					
GC034806	M	5 y	chr22_20043265_20064650	exons 2–9 of *TANGO2*	encodes transport and golgi organization 2 homologs; bi-allelic mutations in TANGO2 cause recurrent muscle weakness with rhabdomyolysis, metabolic crises, and cardiac arrhythmia
**uncertain pathogenicity**					
GC028957	F	27 y	chr15_26914781_26914903	exon 7 of *GABRA5*	encodes γ-aminobutyric acid type A receptor alpha5 subunit; restricted expression toward brain; GABA is the major inhibitory neurotransmitter in the mammalian brain where it acts at GABA-A receptors, which are ligand-gated chloride channels
GC028958	F	2 weeks	chr15_26937134_26937376	exon 8 of *GABRA5*	encodes γ-aminobutyric acid type A receptor alpha5 subunit; restricted expression toward brain; GABA is the major inhibitory neurotransmitter in the mammalian brain where it acts at GABA-A receptors, which are ligand-gated chloride channels
GC030952	F	24 y	chr16_75635620_75635862	exon 7 of *KARS*	encodes lysyl-tRNA synthetase; defects in KARS are associated with the recessive form of Charcot-Mary-Tooth polyneuropathy, the autosomal recessive non-syndromic hearing loss, congenital visual impairment and progressive microcephaly, hypertrophic cardiomyopathy and combined mitochondrial respiratory chain defect
			chr22_26601827_26608074	exons 4–5 of *CRYBB1*	encodes crystallin β B1; mutations in CRYBB1 are associated with congenital cataract with nystagmus
GC035414	M	9 y	Chr22:18906447-18931248	*DGCR6*, *PRODH*	Proline dehydrogenase is involved in the degradation of the amino acid proline. It catalyzes the conversion of proline to pyrroline-5-carboxylate, or P5C. It is associated with Hyperprolinemia, type I, and susceptibility to schizophrenia
GC030953	M	3 y	chr12_21568814_21580552	exons 9–12 of *GYS2*	encodes glycogen synthase 2; mutations in this gene cause autosomal recessive glycogen storage disease type 0 (GSD-0)—a rare type of early childhood fasting hypoglycemia with decreased liver glycogen content
			chr23_55751086_55751208	exon 7 of *RRAGB*	encodes Ras related GTP binding protein B
GC034767	F	3 y	chr1_202135375_202154374	exons 1–5 of *ARL8A*, exons 6–10 of *PTPN7*	ARL8A—encodes ADP ribosylation factor like GTPase 8A; PTPN7—encodes protein tyrosine phosphatase non-receptor type 7
GC034773	M	6 y	chr2_3702340_4228633	exon 12 of *ALLC*, *DCDC2C*, *LINC01304*, ENSG00000215960	ALLC encodes allantoicase, expressed in testis; DCDC2C encodes doublecortin domain containing 2C, low expression in testis; LINC01304 expressed in testis
GC034774	F	35 y	chr12_6452218_6452340	exon 1 of *TAPBPL*	encodes TAP binding protein like, involved in controlling peptide presentation to the immune system
GC034780	M	10 y	chr8_3029293_3230255	exons 28–52 of *CSMD1*	encodes CUB and Sushi multiple domains 1, expressed at intermediate level in brain, including cerebellum, substantia nigra, hippocampus and fetal brain; potential suppressor of squamous cell carcinomas; altered expression of the CSMD1 gene in the peripheral blood of schizophrenia patients; association with cognitive function
GC034789	F	6 m	chr1_145850735_145854625	exons 8–12 of *PIAS3*	encodes protein inhibitor of activated STAT 3, plays a crucial role as a transcriptional coregulation in various cellular pathways, including the STAT pathway and the steroid hormone signaling pathway
			chr1_145868270_145875066	exons 8–11 of *ANKRD35*	encodes ankyrin repeat domain 35, expression in skin, esophagus, association with chronic lymphocytic leukemia
GC034850	M	3 m	chr1_196902403_196914600	exons 2–8 of *CFHR4*	encodes complement factor H related 4, expressed in liver, can associate with lipoproteins and may play a role in lipid metabolism
GC034825	F	34 y	chr14_23988867_23990346	exon 2 of *DHRS4L2*	encodes a member of the short chain dehydrogenase reductase family. The encoded protein may be a NADPH-dependent retinol oxidoreductase

**Table 5 genes-14-00680-t005:** Pathogenic, likely pathogenic, and variants of uncertain significance identified by array and exome sequencing. For the likely pathogenic variants, only CNVs and SNVs which correlated with the patient phenotypes have been included.

	Pathogenic	Likely Pathogenic	Variants of Uncertain Significance
**aCGH** **159 cases**	**CNV + aneuploidy**		
	5 (3.14%)	5 (3.14%)	21 (13.21%)
**ES** **86 cases**	**SNV + small deletions**		
	1 deletion + 2 SNVs 3 (3.49%)	5 SNVs 5 (5.81%)	15 deletions + 911 SNVs

## Data Availability

Any data requests can be directed to the corresponding author.

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
