# Peer review of "Coexisting Conditions Modifying Phenotypes of Patients with 22q11.2 Deletion Syndrome"

_genes, 2023, doi:10.3390/genes14030680_

Round 1
Reviewer 1 Report
This manuscript presents data on additional rare variants identified in 159 cases with a 22q11.2 microdeletion using genome-wide microarray in all, and exome sequencing of a subset of 85 cases. Timing of the additional genotyping was prenatal for 82 cases and “postnatal” at an unspecified age for 77 cases. For some of the variants identified there are some phenotypic data presented.
I believe that these genotyping results are of interest to the research and clinical community. However, the results would be of greater interest, and one could actually appreciate their potential research and clinical merit, if the focus in the manuscript was more clearly on well-defined pathogenic or likely pathogenic variants, and if phenotypic data (including basic demographics like age – even age range or timing of the genotyping if prenatal - and sex) were provided in a consistent manner. To be convincing as to the relevance of their contribution to clinical diagnostics, or even for basic research purposes, it would be essential that the severity of phenotype (or lack thereof) be presented. Other results (e.g., for rare variants of uncertain significance) may be of some interest but would be better placed in supplemental material – again, however, more informative if together with some / any basic phenotypic data.
Taking each section of this very long manuscript in turn:
Abstract – This needs to be rewritten for clarity of purpose, and clarity of actual results. The promise in the abstract is “etiology of clinical variability (in individuals 22q11.2 deletions) between patients with the same deletion size”, and “additional variants (that could) affect the phenotype”, and, “the extension of diagnostics” that could “reveal significant changes in (as many as) 15.6% of patients”. However, but this is not the case with respect to the data on the extent of the 22q11.2 deletions presented, nor with respect to the results on the additional rare variants that are currently presented in the manuscript. If only 85 cases had exome data, then the denominator would appear to be 85 for rare SNVs, and this should be made clear. Even the simple percentages do not appear to be quite accurate – or are unexplained. If there are seven (perhaps 8, see below) cases with additional 9 (perhaps 10) pathogenic or likely pathogenic CNVs, then the percentage of cases is not 5.7%. Other figures of 44.7% and 15.6% are presented without clear attribution.
Introduction – The introduction could be much shorter, and focused on brief background and rationale specific to the current manuscript (vs providing lengthy details). Some belongs (briefly) in Discussion. Again, the promise that the results indicate “in some cases, complex, atypical, as well as common clinical features…may be a consequence of the combined occurrence of the 22q11.2 deletion and a “second hit” pathogenic or likely pathogenic CNV or SNV” does not reflect what is presented in the Results.
Materials and Methods – None of the numbers presented in section 2.1 match those in the Abstract, thus it remains unclear what results are being reported on what sample and assessed by what molecular method (e.g., “20 patients have been analyzed by both methods”). And by what ascertainment method. “Deep phenotyping” is promised (and indeed it is stated that “a detailed form with over 230 clinical features was completed” “for every patient”, but no data are presented in this regard, and it remains unclear for which cases this was (or could be) accomplished, e.g., given that over half of the cases were “prenatal” with unknown fate. In section 2.2, under the heading “aCGH”, first sentence, the term “diagnostic tests” is used with a confusing set of resolution / restriction details that seem to confound exome sequencing done on a research basis (and with questionable utility for CNV detection) with arrayCGH, a rather low-resolution method for CNV detection compared with SNP array methods. Resolution, however, is certainly better than the stated “>200 kb”. It is unclear if any confirmatory / validation methods were used for any rare variant identified, further adding to the uncertainty about results presented. The resolution and limitations of the exome sequencing method used are not presented, including for deletion detection which seems to be a focus (e.g., for the term “small”). Again, it is unclear if any confirmatory / validation methods were used for any rare variant identified. Clinical interpretation of rare variants should follow current ACMG guidelines (or explain why these were not used). There is mention of recurrent CNVs but the results are not restricted to these. It would also be useful to know / report on inheritance status of the 22q11.2 deletions, as known. In the Introduction the authors note the potential importance of this variable.
Results – As above, the results need to be presented in a concise and clear manner. This should start with the 22q11.2 deletion itself, and the numbers with the common A-D vs proximal nested A-B, A-C, vs distal nested B-D, C-D deletion extents. It is possible that, as over half of cases were detected prenatally, that a greater proportion of nested deletions than usually expected may have been detected. If these have somewhat milder phenotypes than for the A-D deletion, then perhaps they may be more likely to have additional rare CNVs that could affect the phenotype. Indeed, this is what appears to be the case in Table 1. Six of the 7 patients had nested 22q11.2 deletions: 4 A-B, 1 A-C, 1 B-D. This may be one of the most interesting findings from the study, but it does not appear to be noted anywhere or discussed. The first case in the table makes one consider a possible translocation, given the 9 Mb 11q24.211q25 (Jacobsen syndrome) deletion and I wonder if this possibility was examined, and if there were any other phenotypic details available than “fetal death”. Some of the CNVs do appear to be extensions of the typical 22q11.2 deletion, either proximally or distally, and one wonders if these possibilities were considered, and if other methods were used to examine these possibilities. The complexity of the region can make some results appear at first using data from algorithms to be discontinuous but on closer inspection are contiguous (e.g., B-F for PD8038, or variants previously reported, see Guo et al., 2018 PMID: 29361080). This is a very important consideration given the authors’ stated interpretation in regard to “additional variants on chromosome 22q”.
Again, the results would be more interpretable and one could appreciate their potential research and clinical merit, if in the manuscript the tables presented, in a consistent manner, only the additional pathogenic and likely pathogenic variants, together with the 22q11.2 deletion extent, and with phenotypic data ideally as “typical” 22q11.2 deletion findings and “possible additional / special / highlighted” findings, together with basic demographics like age (even age range or timing of the genotyping if prenatal) and sex.
If the conclusions (and abstract) are going to emphasize potentially clinically relevant rare CNVs as a group, including those identified by exome sequencing, then these should follow the array results in the manuscript. For example, Figure 1 and Table 5 present a 21.4 kb exonic CNV overlapping TANGO2, that would appear to be of potential clinical interest if phenotype were presented, and there was confirmation using inspection of array data or another method. On the other hand, the known common CNV variant overlapping PRODH and DGCR6 within the LCRA region would appear unlikely to be of clinical importance, or would require far more data to be convincing as deemed here to be “likely pathogenic” (see Guo et al., 2018 PMID: 29361080 for discussion). Consideration of locus, and additional confirmatory data, for the two variants reported overlapping GABRA5 at the complex 15q13 region, and KARS gene results, in Table 5, reported as “likely pathogenic” would also be welcome. Over-interpretation detracts from the main messages of the results of this study.
Table 2 should be in the supplemental material unless there is any convincing phenotypic data that would relate to any of these SNVs within the complex 22q11.2 deletion region (only possible at a glance may be for a CDC45 variant, and even then this is doubtful), most of which are not so rare and none of which have any ACMG clinical variant data or phenotypic data provided. The same appears to be the case for Tables 3 and 4 with respect to genomewide SNVs, with the possible exception of the case with a known NF1 variant but even there the absence of ACMG clinical variant data and phenotype is problematic for interpretation. The authors need to sort the wheat from the chaff and present only the clinically relevant results.
Table 6 (labelled as “Tab”) is unnecessary, and superfluous to the text (when the text is corrected). If 100% (which appears likely, and expected) of individuals had rare SNVs detected of those assessed using exome sequencing, the issue is interpretation, and this should be stated somewhere, likely in the discussion.
Discussion – This needs to be made far clearer, cut by about half, and relate *only* to the findings from the study results presented (not from the authors’ previous studies or details of others’ studies that are unrelated). There is no hypothesis tested in this study. There should be no results presented in the Discussion that are not summarized in the Results (e.g., duplications). This study does not demonstrate that “secondary diagnoses should be taken into account especially when less common 22q11.2DS features are present” – this appears to be a conclusion for a different study. The limitations section, on the other hand, needs to be expanded.
Other issues – The are basic accuracy, terminology, (i.e., scientific) and English language issues to be addressed throughout the manuscript, that require careful proof-reading.
Author Response
To: Editor-in-Chief, Genes
14 February, 2023
Dear Editor,
We thank the Reviewers for their comments and suggestions and are pleased to submit a revised version of our manuscript, “Coexisting conditions modifying phenotypes of patients with 22q11.2 Deletion Syndrome” for publication in Genes. We outline the changes made to the manuscript below the comments highlighted in yellow.
In addition, all changes to the main text are also marked in yellow.
Reviewer 1
Comments and Suggestions for Authors
This manuscript presents data on additional rare variants identified in 159 cases with a 22q11.2 microdeletion using genome-wide microarray in all, and exome sequencing of a subset of 85 cases. Timing of the additional genotyping was prenatal for 82 cases and “postnatal” at an unspecified age for 77 cases. For some of the variants identified there are some phenotypic data presented.
I believe that these genotyping results are of interest to the research and clinical community. However, the results would be of greater interest, and one could actually appreciate their potential research and clinical merit, if the focus in the manuscript was more clearly on well-defined pathogenic or likely pathogenic variants, and if phenotypic data (including basic demographics like age – even age range or timing of the genotyping if prenatal - and sex) were provided in a consistent manner. To be convincing as to the relevance of their contribution to clinical diagnostics, or even for basic research purposes, it would be essential that the severity of phenotype (or lack thereof) be presented.
We have added supplementary table with the clinical features, size of the 22q11.2 deletion, sex and age of the patients underwent ES procedure (Supplementary Table S2). Additionally, we have added information about patients` age and sex in all tables.
Other results (e.g., for rare variants of uncertain significance) may be of some interest but would be better placed in supplemental material – again, however, more informative if together with some / any basic phenotypic data.
The table with the rare variants of uncertain significance has been moved to the Supplementary materials (Supplementary Table S3).
Taking each section of this very long manuscript in turn:
Abstract – This needs to be rewritten for clarity of purpose, and clarity of actual results. The promise in the abstract is “etiology of clinical variability (in individuals 22q11.2 deletions) between patients with the same deletion size”, and “additional variants (that could) affect the phenotype”, and, “the extension of diagnostics” that could “reveal significant changes in (as many as) 15.6% of patients”.
Those sentences have been removed:
The aim of our study was to investigate the etiology of clinical variability between patients with the recurrent deletion. The alterations in the other part of genome or in the remaining 22q11 allele is one of the hypotheses that might explain part of this clinical variability.
The aim of study was changed to:
The aim of our study was to investigate how often the additional variants in the genome, can affect clinical variation, among patients with the same deletion size. To examine the presence of additional variants, affecting the phenotype we performed microarray in 82 prenatal and 77 postnatal cases and performed exome sequencing in 86 postnatal patients with 22q11.2DS.
However, but this is not the case with respect to the data on the extent of the 22q11.2 deletions presented, nor with respect to the results on the additional rare variants that are currently presented in the manuscript. If only 85 cases had exome data, then the denominator would appear to be 85 for rare SNVs, and this should be made clear. Even the simple percentages do not appear to be quite accurate – or are unexplained. If there are seven (perhaps 8, see below) cases with additional 9 (perhaps 10) pathogenic or likely pathogenic CNVs, then the percentage of cases is not 5.7%. Other figures of 44.7% and 15.6% are presented without clear attribution.
Has been changed to:
Within those 159 patients where array was performed, 5 pathogenic and 5 likely pathogenic CNVs have been identified, outside of the 22q11.2 region. This indicates that in 6.28 % cases, additional CNVs most likely contribute to the clinical presentation.
Introduction – The introduction could be much shorter, and focused on brief background and rationale specific to the current manuscript (vs providing lengthy details). Some belongs (briefly) in Discussion. Again, the promise that the results indicate “in some cases, complex, atypical, as well as common clinical features…may be a consequence of the combined occurrence of the 22q11.2 deletion and a “second hit” pathogenic or likely pathogenic CNV or SNV” does not reflect what is presented in the Results.
The Introduction section has been shortened, as suggested.
Materials and Methods – None of the numbers presented in section 2.1 match those in the Abstract, thus it remains unclear what results are being reported on what sample and assessed by what molecular method (e.g., “20 patients have been analyzed by both methods”). And by what ascertainment method.
As described in the Material and Method section, we have changed to:
During routine aCGH diagnostics at the Institute of Mother and Child, we have identified 159 patients with 22q11.2 deletion (82 prenatal and 77 postnatal). For exome sequencing, 86 postnatal patients (40 females and 46 males) were recruited/evaluated at the Department of Medical Genetics, Institute of Mother and Child, Warsaw, Poland (76 probands) and within the 22q and You Center, The Children's Hospital of Phila-delphia, Philadelphia, PA, USA (10 probands). 20 patients have been analyzed by both methods; array CGH and exome sequencing. In remaining 66 patients, the presence of a chromosome 22q11.2 deletion had been identified by FISH or MLPA.
Additionally, in each table we have included ID of the patients.
“Deep phenotyping” is promised (and indeed it is stated that “a detailed form with over 230 clinical features was completed” “for every patient”, but no data are presented in this regard, and it remains unclear for which cases this was (or could be) accomplished, e.g., given that over half of the cases were “prenatal” with unknown fate.
In the supplementary data we have added Table S2, with all the clinical information for patients where exome sequencing was performed. For prenatal cases were we have found the additional CNV, clinical data was provided in Table 1.
In section 2.2, under the heading “aCGH”, first sentence, the term “diagnostic tests” is used with a confusing set of resolution / restriction details that seem to confound exome sequencing done on a research basis (and with questionable utility for CNV detection) with arrayCGH, a rather low-resolution method for CNV detection compared with SNP array methods. Resolution, however, is certainly better than the stated “>200 kb”. It is unclear if any confirmatory / validation methods were used for any rare variant identified, further adding to the uncertainty about results presented. The resolution and limitations of the exome sequencing method used are not presented, including for deletion detection which seems to be a focus (e.g., for the term “small”). Again, it is unclear if any confirmatory / validation methods were used for any rare variant identified. Clinical interpretation of rare variants should follow current ACMG guidelines (or explain why these were not used). There is mention of recurrent CNVs but the results are not restricted to these. It would also be useful to know / report on inheritance status of the 22q11.2 deletions, as known. In the Introduction the authors note the potential importance of this variable.
The paragraph from section 2.2 aCGH:
Diagnostic tests and verification of predicted large (>200 Kb in size) and rare (<0.1% frequency) CNVs, identified by exome sequencing, were performed by using array comparative genomic hybridization (arrayCGH) with commercially available arrays the 60K CytoSure Constitutional v3 microarray (Oxford Gene Technology, Oxford, UK) according to manufacturer’s instructions. CNVs that showed partial or complete overlap with known segmental duplications were excluded from further analysis, as these regions are naturally susceptible to copy number changes and are unlikely to cause monogenic disease. CNVs were classified as pathogenic, likely pathogenic, likely benign, benign and of uncertain significance (VUS) based on clinical data in known CNV databases: Database of Genomic Variants (http://dgv.tcag.ca/dgv/app/home), ClinVar (https://www.ncbi.nlm.nih.gov/clinvar/), DECIPHER (https://www.deciphergenomics.org/), ClinGen (https://www.clinicalgenome.org/). Recurrent CNVs or CNVs associated with known microdeletion or microduplication syndromes were classified as pathogenic or likely pathogenic depending on the penetrance and clinical features present in probands.
Has been changed to:
We performed array comparative genomic hybridization (arrayCGH), with commercially available arrays the 60K CytoSure Constitutional v3 microarray (Oxford Gene Technology, Oxford, UK) according to manufacturer’s instructions. CNVs that showed partial or complete overlap with known segmental duplications were excluded from further analysis, due to the high variability of copy number variations in those regions. CNVs were classified as pathogenic, likely pathogenic, likely benign, benign and of uncertain significance (VUS) based on clinical data in known CNV databases: Database of Genomic Variants (http://dgv.tcag.ca/dgv/app/home), ClinVar (https://www.ncbi.nlm.nih.gov/clinvar/), DECIPHER (https://www.deciphergenomics.org/), ClinGen (https://www.clinicalgenome.org/). Recurrent CNVs or CNVs associated with known microdeletion or microduplication syndromes were classified as pathogenic or likely pathogenic depending on the penetrance and clinical features present in probands. In 159 patients (82 prenatal and 77 postnatal) array CGH was performed as a first diagnostic test. Additionally, we conducted aCGH where exome sequencing revealed CNV, which could be confirmed (due to size and array coverage).
Results – As above, the results need to be presented in a concise and clear manner. This should start with the 22q11.2 deletion itself, and the numbers with the common A-D vs proximal nested A-B, A-C, vs distal nested B-D, C-D deletion extents. It is possible that, as over half of cases were detected prenatally, that a greater proportion of nested deletions than usually expected may have been detected. If these have somewhat milder phenotypes than for the A-D deletion, then perhaps they may be more likely to have additional rare CNVs that could affect the phenotype. Indeed, this is what appears to be the case in Table 1. Six of the 7 patients had nested 22q11.2 deletions: 4 A-B, 1 A-C, 1 B-D. This may be one of the most interesting findings from the study, but it does not appear to be noted anywhere or discussed.
We added in the Discussion:
What is particularly interesting, additional CNVs occurred in 7 patients with nested 22q11 deletions and 14 patients with A-D typical deletion. Taking into account the ra-tio of nested to the typical deletions (28:131 in our study), the result can suggest, that if the smaller deletions have milder phenotype, than the classical ones, they may have more likely additional CNVs affecting phenotype. However, to draw the final conclu-sions, more cases with different sizes of the deletion need to be studied.
The first case in the table makes one consider a possible translocation, given the 9 Mb 11q24.211q25 (Jacobsen syndrome) deletion and I wonder if this possibility was examined, and if there were any other phenotypic details available than “fetal death”.
We have added the available clinical features to the table:
lip and cleft palate, fetal edema
The parents refused further testing.
Some of the CNVs do appear to be extensions of the typical 22q11.2 deletion, either proximally or distally, and one wonders if these possibilities were considered, and if other methods were used to examine these possibilities. The complexity of the region can make some results appear at first using data from algorithms to be discontinuous but on closer inspection are contiguous (e.g., B-F for PD8038, or variants previously reported, see Guo et al., 2018 PMID: 29361080). This is a very important consideration given the authors’ stated interpretation in regard to “additional variants on chromosome 22q”.
In the Supplementary Table S2 we added the coordinates where the size of the deletion was extended.
In the Results and Discussion we also added:
In 10 cases the additional CNV was located on chromosome 22q11, however in 7 of those cases the additional deleted regions most likely were the extension of the typical or nested 22q11 deletions.
Again, the results would be more interpretable and one could appreciate their potential research and clinical merit, if in the manuscript the tables presented, in a consistent manner, only the additional pathogenic and likely pathogenic variants, together with the 22q11.2 deletion extent, and with phenotypic data ideally as “typical” 22q11.2 deletion findings and “possible additional / special / highlighted” findings, together with basic demographics like age (even age range or timing of the genotyping if prenatal) and sex.
The age and sex of the patients has been added into the Table 1-4. The size of the deletion has also been added into the Supplementary Table 2 where all the know clinical features are listed. The Table 2 with variants of uncertain clinical significance has been moved to the supplementary data. However, we think it is important leaving the VOU variants, that may change their classification in the future, and that may be of interest for further studies.
If the conclusions (and abstract) are going to emphasize potentially clinically relevant rare CNVs as a group, including those identified by exome sequencing, then these should follow the array results in the manuscript. For example, Figure 1 and Table 5 present a 21.4 kb exonic CNV overlapping TANGO2, that would appear to be of potential clinical interest if phenotype were presented, and there was confirmation using inspection of array data or another method.
We confirmed the deletion by qPCR method. The information was added into the Result section.
On the other hand, the known common CNV variant overlapping PRODH and DGCR6 within the LCRA region would appear unlikely to be of clinical importance, or would require far more data to be convincing as deemed here to be “likely pathogenic” (see Guo et al., 2018 PMID: 29361080 for discussion). Consideration of locus, and additional confirmatory data, for the two variants reported overlapping GABRA5 at the complex 15q13 region, and KARS gene results, in Table 5, reported as “likely pathogenic” would also be welcome. Over-interpretation detracts from the main messages of the results of this study.
We have changed the classification of all the previously classified as “likely pathogenic” variants into “uncertain pathogenicity”
Table 2 should be in the supplemental material unless there is any convincing phenotypic data that would relate to any of these SNVs within the complex 22q11.2 deletion region (only possible at a glance may be for a CDC45 variant, and even then this is doubtful), most of which are not so rare and none of which have any ACMG clinical variant data or phenotypic data provided.
Table 2 has been removed from the main text and included in the supplementary data as Table S3.
The same appears to be the case for Tables 3 and 4 with respect to genomewide SNVs, with the possible exception of the case with a known NF1 variant but even there the absence of ACMG clinical variant data and phenotype is problematic for interpretation. The authors need to sort the wheat from the chaff and present only the clinically relevant results.
We have added:
For two cases the clinical features are consistent with the predicted consequences of genomic variant and has been classified as pathogenic. In five cases, the patients did not present expected phenotype features, therefore the variants has been classified as potentially pathogenic.
Additionally, in table 3 the new column has been added with “Clinical features associated with variant”
Table 6 (labelled as “Tab”) is unnecessary, and superfluous to the text (when the text is corrected). If 100% (which appears likely, and expected) of individuals had rare SNVs detected of those assessed using exome sequencing, the issue is interpretation, and this should be stated somewhere, likely in the discussion.
We corrected the Table 6 (now Table 5) and we hope it gives the more clarification to our results.
Discussion – This needs to be made far clearer, cut by about half, and relate *only* to the findings from the study results presented (not from the authors’ previous studies or details of others’ studies that are unrelated). There is no hypothesis tested in this study. There should be no results presented in the Discussion that are not summarized in the Results (e.g., duplications). This study does not demonstrate that “secondary diagnoses should be taken into account especially when less common 22q11.2DS features are present” – this appears to be a conclusion for a different study. The limitations section, on the other hand, needs to be expanded.
We have shorten the Discussion section and focused more on the obtained results.
Other issues – The are basic accuracy, terminology, (i.e., scientific) and English language issues to be addressed throughout the manuscript, that require careful proof-reading.
We have corrected the language and terminology where errors have been noticed. We also asked the native speaker to correct the manuscript.
Reviewer 2
Comments and Suggestions for Authors
Review for Genes
22q11.2 deletion syndrome results in a spectrum of clinical phenotypes. This variability occurs despite most patients having similar sized chromosomal deletions on 22q11.2. The authors of the current submission assess the possible contribution of additional genetic variants among a cohort of 159 pre- and post-natal 22q11.2 deletion syndrome patients. This is done by a combination of microarray analyses (all 159 individuals) and for 85, whole exome sequencing. Several interesting findings are reported, with information separated into 4 Tables, using CNVs and SNVs differences as indicators. Using such strategies, they noted that 25% of the 22q patients had mutations in the second retained allele of 22q11.2. The results are of importance for clinicians seeking to understand the complexity of 22q11.2 deletion syndrome and how outlying mutations could affect the phenotype. This later point remains one of the weaknesses of the paper, as it remains unclear whether any of the genes impacted on the remaining 22q11.2 allele and/or other chromosomal regions correlate with the severity and penetrance of the clinical spectrum of the 22q11.2 deletion syndrome phenotypes. This should at the very least, be discussed. Also, the way the data is presented could be improved to make this manuscript of value to readers. These concerns and other issues are listed next.
Comments
- Line 32. In the abstract, the authors states “This indicates that in 5.7% cases, additional CNVs most likely contribute to the clinical presentation”. This has not been proven for most of the cases, so such a statement is too strong unless supporting data is included.
The Abstract has been changed:
Within those 159 patients where array was performed, 5 pathogenic and 5 likely pathogenic CNVs have been identified, outside of the 22q11.2 region. This indicates that in 6.28 % cases, additional CNVs most likely contribute to the clinical presentation. Additionally, exome sequencing in 86 patients revealed 3 pathogenic (3.49%) and 5 likely pathogenic (5.81%) SNVs and small CNV.
- In the introduction, the authors comment that no single gene dose difference can account for the phenotypic variability in 22q11.2 deletion. However, a statement that the mechanism of Tbx1 function (haploinsufficient) could account for some of the phenotypic variability is warranted. This is because Tbx1 is a major contributor to the congenital malformations, and the variability could be explained by the limited and variable marking of DNA. This could create more random epigenetic marking, based on the role of Tbx1 in interacting with histone methyltransferases (Kmct).
We agree with the comments, and we added the information about the TBX1 gene:
It remains unknown why individuals with deletion of the same size, present such a wide range of phenotypes. Some of the genes located within the standard deletion have major clinical impact, particularly T-box 1 (TBX1). The TBX1 gene is part of the larger family of T-box genes, which help to regulate tissue and organ formation during development. However, a different minority of patients harbor nested distal deletions but retain two copies of the TBX1 gene [2]. Therefore, the single dose of deleted genes alone cannot explain the tremendous variation in the severity and penetrance of the associated clinical features among affected individuals.
However, the suggested mechanism (epigenetics), which is of course possible is beyond our work and explaining this in the Introduction section will be out of scope of the rest of the manuscript and irrelevant to our research.
- Line 185. Table 1. The listed phenotypes for the patient should include 1 column that lists all 22q11.2 syndromes for the patient along with a second column that lists new phenotypes not common to 22q.11. A third new column wherein an impacted gene is potentially coupled to overlapping conditions (e.g., heart defects, hypoparathyroidism, etc.) would enable the reader to link mutations to phenotypic variability/severity in the patient cohort.
We have tried to modify the Table according to the suggestions. Unfortunately, it is almost impossible to distinguish the clinical features typical and not typical for patients with 22q11. Even those which are less typical, we could not assign to the genes in other aberrations. Also, not for all patients the full phenotype was available.
- Line 207. Table 2. The authors mention that 25% of patients had mutations on the undeleted allele of 22q11.2. Non-synonymous differences and frameshifts levels are very high. Is such a high frequency noted in the non-22q11.2 population of individuals or is this unique to 22q11.2 deletion syndrome. If unique, what could explain such a pattern?
We have compare the 22q11 cohort to the patients cohort without 22q11 deletion, sequenced in our Department. The level of variants is similar. However, taking into the consideration, that in patients with 22DS we have only one allele, the cut-off of frequency of the variants is higher than we would analyze in cohort without deletion. We did not analyze patients’ cohort of other deletion syndrome.
- Line 246. Table 3. The authors should link the information in the Table to the severity and penetrance of the 22q11.2 deletion syndrome phenotypes reported for each patient designated in the Table. Table between Lines 249-250. Is the OMIM disease that is listed reported for the patient or is this just what the gene mutation has been linked to in the literature. This was done for Table 4 and would be informative if done for Table 3. It would also be informative to describe the extent of the deletion noted for each patient listed in each of the Tables.
We have added the column called: “Clinical features associated with variant”
Additionally in the main text we have added:
For two cases the clinical features are consistent with the predicted consequences of genomic variant and has been classified as pathogenic. In five cases, the patients did not present expected phenotype features, therefore the variants has been classified as potentially pathogenic.
We have added supplementary table with the clinical features, size of the 22q11.2 deletion, sex and age of the patients underwent ES procedure (Supplementary Table S2). Additionally, we have added information about patients` age and sex in all tables.
- Line 276. “Pathogenic, likely pathogenic and deletions of undertrained clinical significance identified by HMZDelFinder”. What does undertrained mean?
It was an typographical error – it should be “uncertain”
Changed to:
Table 4. Pathogenic and of uncertain clinical significance deletions identified by HMZDelFinder
- Line 295. Beginning of the conclusion. The sentence beginning this section is probably a sentence from a previous reviewer related to a prior submission of this manuscript as I cannot see how the author would write “Authors should discuss the results and how they can be interpreted from the per”.
The sentence was added by the Editor at the Journal, and was removed.
- Line 304-324 – it remains unclear from this entire section whether the authors are describing prior published work, from others, or detailing information from the current study. Perhaps a better way to write this section is to mention what has been uncovered in the current study, how this is consistent with some earlier reports and then highlight what has been gained from the analyses undertaken in the current study. For example, are the number of affected individuals higher than previously published. How has the current study gone beyond earlier work? What has been learned?
We have re-written the Discussion section. In all topics we tried to relate our results to the literature data.
- Supplemental Table 1. As described, it sounds like the authors have documented all the clinically relevant information for each patient. This would have been very informative and should have been done. But what is provided is only a template that lists the questions asked for each patient. Also, it remains unclear for the pre-natal cohort, how many survived and whether they had more severe phenotypes than the post-natal that caused death is some. It is also unclear whether the pre-natal cohort were processed as postmortem. Again, important information that can help related genetic variations to disease severity.
We have added supplementary table with the clinical features, size of the 22q11.2 deletion, sex and age of the patients underwent ES procedure (Supplementary Table S2).
For the pre-natal cases we do not have these information. The patients are from different centers in Poland and very often after diagnosis, the medical care of the woman and her baby is monitored by other clinical centers. We have added in the Table 1 the information about the week of pregnancy.
- Proofread and run spell check for the entire document. Errors do exist. e.g., line 194 remining written instead of remaining in multiple locations.
We have changed the :
remining
to:
Remaining
We have corrected the language and terminology where errors have been noticed. We also asked the native speaker to correct the manuscript. Additionally all gene names were changed to “italic” font.
Reviewer 2 Report
Review for Genes
22q11.2 deletion syndrome results in a spectrum of clinical phenotypes. This variability occurs despite most patients having similar sized chromosomal deletions on 22q11.2. The authors of the current submission assess the possible contribution of additional genetic variants among a cohort of 159 pre- and post-natal 22q11.2 deletion syndrome patients. This is done by a combination of microarray analyses (all 159 individuals) and for 85, whole exome sequencing. Several interesting findings are reported, with information separated into 4 Tables, using CNVs and SNVs differences as indicators. Using such strategies, they noted that 25% of the 22q patients had mutations in the second retained allele of 22q11.2. The results are of importance for clinicians seeking to understand the complexity of 22q11.2 deletion syndrome and how outlying mutations could affect the phenotype. This later point remains one of the weaknesses of the paper, as it remains unclear whether any of the genes impacted on the remaining 22q11.2 allele and/or other chromosomal regions correlate with the severity and penetrance of the clinical spectrum of the 22q11.2 deletion syndrome phenotypes. This should at the very least, be discussed. Also, the way the data is presented could be improved to make this manuscript of value to readers. These concerns and other issues are listed next.
Comments
1. Line 32. In the abstract, the authors states “This indicates that in 5.7% cases, additional CNVs most likely contribute to the clinical presentation”. This has not been proven for most of the cases, so such a statement is too strong unless supporting data is included.
2. In the introduction, the authors comment that no single gene dose difference can account for the phenotypic variability in 22q11.2 deletion. However, a statement that the mechanism of Tbx1 function (haploinsufficient) could account for some of the phenotypic variability is warranted. This is because Tbx1 is a major contributor to the congenital malformations, and the variability could be explained by the limited and variable marking of DNA. This could create more random epigenetic marking, based on the role of Tbx1 in interacting with histone methyltransferases (Kmct).
3. Line 185. Table 1. The listed phenotypes for the patient should include 1 column that lists all 22q11.2 syndromes for the patient along with a second column that lists new phenotypes not common to 22q.11. A third new column wherein an impacted gene is potentially coupled to overlapping conditions (e.g., heart defects, hypoparathyroidism, etc.) would enable the reader to link mutations to phenotypic variability/severity in the patient cohort.
4. Line 207. Table 2. The authors mention that 25% of patients had mutations on the undeleted allele of 22q11.2. Non-synonymous differences and frameshifts levels are very high. Is such a high frequency noted in the non-22q11.2 population of individuals or is this unique to 22q11.2 deletion syndrome. If unique, what could explain such a pattern?
5. Line 246. Table 3. The authors should link the information in the Table to the severity and penetrance of the 22q11.2 deletion syndrome phenotypes reported for each patient designated in the Table. Table between Lines 249-250. Is the OMIM disease that is listed reported for the patient or is this just what the gene mutation has been linked to in the literature. This was done for Table 4 and would be informative if done for Table 3. It would also be informative to describe the extent of the deletion noted for each patient listed in each of the Tables.
6. Line 276. “Pathogenic, likely pathogenic and deletions of undertrained clinical significance identified by HMZDelFinder”. What does undertrained mean?
7. Line 295. Beginning of the conclusion. The sentence beginning this section is probably a sentence from a previous reviewer related to a prior submission of this manuscript as I cannot see how the author would write “Authors should discuss the results and how they can be interpreted from the per”.
8. Line 304-324 – it remains unclear from this entire section whether the authors are describing prior published work, from others, or detailing information from the current study. Perhaps a better way to write this section is to mention what has been uncovered in the current study, how this is consistent with some earlier reports and then highlight what has been gained from the analyses undertaken in the current study. For example, are the number of affected individuals higher than previously published. How has the current study gone beyond earlier work? What has been learned?
9. Supplemental Table 1. As described, it sounds like the authors have documented all the clinically relevant information for each patient. This would have been very informative and should have been done. But what is provided is only a template that lists the questions asked for each patient. Also, it remains unclear for the pre-natal cohort, how many survived and whether they had more severe phenotypes than the post-natal that caused death is some. It is also unclear whether the pre-natal cohort were processed as postmortem. Again, important information that can help related genetic variations to disease severity.
10. Proofread and run spell check for the entire document. Errors do exist. e.g., line 194 remining written instead of remaining in multiple locations.
Author Response

(The authors gave the same response as above.)
